

# Anthropometric, physical function and general health markers of Masters athletes: a cross-sectional study

Samantha Fien[1], Mike Climstein[2,3], Clodagh Quilter[1,4], Georgina Buckley[1,4], Timothy Henwood[1,5,6], Josie Grigg[1] and Justin W.L. Keogh[1,7,8]

[1] Health Science and Medicine, Bond University, Robina, Queensland, Australia
[2] Water Based Research Unit, Faculty of Health Sciences, Bond University, Robina, Queensland, Australia
[3] Exercise Health & Performance Faculty Research Group, The University of Sydney, Lidcombe, New South Wales, Australia
[4] Physical Education and Sport Sciences, University of Limerick, Limerick, Ireland
[5] School of Human Movement and Nutritional Science, University of Queensland, Brisbane, Queensland, Australia
[6] Community Wellness and Lifestyle, Southern Cross Care (SA & NT) Inc., Adelaide, South Australia, Australia
[7] Human Potential Centre, Auckland University of Technology, Auckland, New Zealand
[8] Cluster for Health Improvement, Faculty of Science, Health, Education and Engineering, University of the Sunshine Coast, Sunshine Coast, Queensland, Australia

Corresponding author
Samantha Fien, samantha.fien@student.bond.edu.au

## ABSTRACT

Once the general decline in muscle mass, muscle strength and physical performance falls below specific thresholds, the middle aged or older adult will be diagnosed as having sarcopenia (a loss of skeletal muscle mass and strength). Sarcopenia contributes to a range of adverse events in older age including disability, hospitalisation, institutionalisation and falls. One potentially relevant but understudied population for sarcopenia researchers would be Masters athletes. Masters sport is becoming more common as it allows athletes (typically 40 years and older) the opportunity to participate in individual and/or team sports against individuals of similar age. This study examined a variety of measures of anthropometric, physical function and general health markers in the male and female Masters athletes who competed at the 2014 Pan Pacific Masters Games held on the Gold Coast, Australia. Bioelectrical impedance analysis was used to collect body fat percentage, fat mass and fat-free mass; with body mass, height, body mass index (BMI) and sarcopenic status also recorded. Physical function was quantified by handgrip strength and habitual walking speed; with general health described by the number of chronic diseases and prescribed medications. Between group analyses utilised ANOVA and Tukey's post-hoc tests to examine the effect of age group (40–49, 50–59, 60–69 and >70 years old) on the outcome measures for the entire sample as well as the male and female sub-groups. A total of 156 athletes (78 male, 78 female; mean 55.7 years) provided informed consent to participate in this study. These athletes possessed substantially better anthropometric, physical function and general health characteristics than the literature for their less physically active age-matched peers. No Masters athletes were categorised as being sarcopenic, although one participant had below normal physical performance and six participants had below normal muscle strength. In contrast, significant age-related reductions in handgrip strength and increases in the number of chronic diseases and prescribed medications were observed for the overall cohort as well as the male and female sub-groups. Nevertheless, even those aged over 70

years only averaged one chronic disease and one prescribed medication. These results may suggest that participation in Masters sport helps to maintain anthropometry, physical function and general health in middle-aged and older adults. However, it is also possible that only healthier middle-aged and older adults with favourable body composition and physical function may be able to participate in Masters sport. Future research should therefore utilise longitudinal research designs to determine the health and functional benefits of Master sports participation for middle-aged and older adults.

# INTRODUCTION

The field of gerontological research has sought to characterize the rate of age-related decline in physical function and overall health and wellness. For example, it has been found that in sedentary individuals $VO_2$max decreases by 10% per decade after the age of 30 years (*Lepers & Stapley, 2016*). The ageing process is also commonly associated with declines in muscle mass and strength that subsequently lead to functional declines, physical disability and falls (*Wroblewski et al., 2011*). This age-related decline in muscle mass and physical function is termed sarcopenia. According to the European Working Group for Sarcopenia in Older People (EWGSOP), sarcopenia has been defined as '*the loss of skeletal muscle mass and strength that occurs with advanced aging*' and represents a diminished state of health (*Cruz-Jentoft et al., 2010*). The diagnosis of sarcopenia is based upon an individual's muscle mass, muscular strength and physical performance. While the EWGSOP endorsed a variety of assessments of muscle mass, strength and physical performance, bioelectrical impedance analysis (*Avery & Lyman, 1867*), handgrip strength and gait speed are valid assessments that are commonly used during field based testing (*Cruz-Jentoft et al., 2010*).

A recent systematic review of studies using the EWGSOP sarcopenia diagnostic criteria has demonstrated sarcopenia prevalence rates of 1–29% in community dwelling adults over the age of 50 years, 14–33% in older adults in long-term care and 10% in acute hospital-care older populations (*Cruz-Jentoft et al., 2014*). Such results suggest that the rate of sarcopenia increases with age and the degree of aged care, with data from our group reporting sarcopenia prevalence rates of ∼40% in Australian aged care settings (*Senior et al., 2015*). It is perhaps even more alarming that residential aged care facilities may have up to 85% of residents below normal thresholds for muscle strength and 97% below the threshold for physical performance (*Keogh et al., 2015*). The increased prevalence of sarcopenic handgrip strength and gait speed is consistent with the study of 415 older adults between the age of 60–99 years, in which the age-related decline in handgrip strength and gait speed was greater than for muscle mass (*Bai et al., 2016*). There is also a general age-related increase in the number of chronic diseases and prescribed medications, with these again often being greater in residential aged care than community dwelling older adults. Specifically, Australian residential aged care adults have been reported to have an

average of 14–15 chronic diseases and 11–15 prescribed medications (*Fien et al., 2016*; *Keogh et al., 2015*).

One segment of the middle-aged and older community dwelling community that would appear of interest to sarcopenia researchers are Masters athletes (*McKean, Manson & Stanish, 2006*; *Pantoja et al., 2016*; *Ransdell, Vener & Huberty, 2009*; *Wroblewski et al., 2011*). Masters athletes are typically individuals aged 40 years or older who systematically train and compete in one or more individual or team sports (*Lepers & Stapley, 2016*). With an ageing of the population and the growing interest in prolonging health and wellbeing there has been a rise in the number of Masters athletes. Recent data for the New York marathon indicating that more than 50% of the total male finishers and more than 40% of female finishers were Masters athletes (*Lepers & Stapley, 2016*). As some Master athletes may have extensive training backgrounds for many decades, they may be an ideal population for gerontological and sarcopenic research. Given their continued levels of sports and activity participation, it would be hypothesised that Masters athletes would have better health than their same aged, but less active peers. To this end, Masters athletes may allow researchers to better understand the true magnitude of the age-related decline in physical function (*Geard et al., in press*; *Young et al., 2008*).

Typically, studies of Masters athletes demonstrate age-related decline in anthropometry, physical function and general health markers, but these declines are less pronounced and typically occur at an older age than in the wider population (*Borges et al., 2016*). For example, in a cross sectional study of 40 high level Masters athletes (20 males and 20 female, aged 40–81 years) who trained 4–5 times week, age was significantly positively correlated to body mass index (BMI) and body fat percentage (*Wroblewski et al., 2011*). When all 40 participants were pooled together, significant declines in quadriceps peak torque were only observed for the participants 60 years and older (*Wroblewski et al., 2011*). Masters athletes are typically individuals aged 40 years or older who systematically train and compete in one or more individual or team sports (*Lepers & Stapley, 2016*). The qualifying age of masters athletes depends upon the sport and may also be dependent upon the specific distance of that sport. For example, the USA Track and Field association defines a Masters athlete as 30 years of age however for longer distances, 40 years of age. The major results of this study was that 30–35 and 35–45 year old groups were not significantly different across any of the anaerobic parameters; with significant decrements only being observed in those aged over 45 years. It was also observed that the age-related decline in muscular power (26–32%) for athletes 45 years and older was greater as compared to the decline seen in muscular strength (16–23%) or velocity (13–14%). Body mass index has also been used as an indicator of general health in several studies of Masters athletes (*Climstein et al., in press*; *Walsh et al., 2011*; *Walsh et al., 2013b*). The largest of these studies examined the BMI of 6,071 master's athletes (51.9% male and 48.1% female) aged 25–91 years of age competing in the Sydney World Masters Games in 2009 (*Walsh et al., 2011*). While a significant positive correlation was found between age and BMI, Masters athletes demonstrated significantly lower BMI than age-matched controls (*Walsh et al., 2011*).

While a number of studies have described the BMI and performance of Masters athletes, little is known about Masters Athletes wider anthropometric, functional and general

 

health characteristics; how these factors may change with ageing and how they compare to age-matched, community dwelling controls. The aims of this cross-sectional study were to: (1) characterize the body composition, muscle strength, physical function, number of chronic diseases and medications in Masters athletes; and (2) gain some insight into how these factors might be influenced by the age and sex of the Masters athletes. It was hypothesized that: (1) Masters athletes across multiple age groups will exhibit high levels of muscle mass, handgrip strength, gait speed and lower levels of chronic disease and prescribed medication usage; and (2) these outcomes will be significantly greater in male than female and younger rather than older Masters athletes. Masters athletes in this context may provide a suitable research model to look at the true physiological changes associated with age.

## METHODS

### Study design

This research utilized cross-sectional observational study design to characterize the body composition, muscle strength, physical function and overall health and a number of health related indicies of Masters athletes competing at the 2014 Pan Pacific Masters Games, Gold Coast, Australia. This study received approval by the Bond University Human Research Ethics Committee (RO1823) in accordance with the ethical standards of the Helsinki Declaration of 1975 (revised in 2008).

Participants were recruited at the Masters Pan Pacific Games whereby participants had to go to registration desk at the Gold Coast Convention Centre and there were a number of stalls in relation to health, nutrition and physical activity. We had a selected area whereby participants could partake if they wanted to.

### Sample

Eligible participants for this study included all competitors (national and international) who were participating in at least one sport (out of 43 sports) available at the 2014 Pan Pacific Masters Games. Following attainment of informed consent (verbal and written) to participate all participants initially completed a brief questionnaire, which included basic demographics (age, sport(s), and training time) and a medical health history questionnaire (chronic disease(s) and prescribed medications).

### Anthropometry

Height was assessed using a stadiometer to an accuracy of 0.5 cm. Body mass and body composition was assessed using a Tanita bioelectrical impedance (BIA) analysis body composition analyzer (Model MC-980MA, Illinois, USA) (*Kelly & Metcalfe, 2012*). The BIA analyzer was calibrated each morning prior to any assessments. Participants were to have avoided strenuous exercise for the previous 24 h, bladder void and be hydrated prior to body composition assessment. The subjects were instructed to remove footwear, step onto the BIA machine and follow the directions of the machine whilst holding the electrodes in each hand. The BIA uses an algorithm based on the relative electrical resistance in lean tissue, fat and water to calculate body composition measures. The Tanita uses four
reference points in the body, hands and both feet and then produces a complete report of fat and muscle percentage. The Australian general population data was used (aged 35 years and older) available from the 2011–2012 Australian Bureau of Statistics Australian Health Survey were used for comparative purposes (*Australian Bureau of Statistics, 2012*).

The anthropometric data collected allowed for each participant to be screened for sarcopenia according to EWGSOP guidelines (*Cruz-Jentoft et al., 2010*). The EWGSOP screening algorithm consisted of assessing gait speed, handgrip strength and muscle mass (i.e., lean mass). In order to be defined sarcopenic, individuals needed to have below normal levels of muscle mass and muscle strength or performance, using gender-specific cut-off points summarised by *Cruz-Jentoft et al. (2010)*.

### Handgrip strength

Handgrip strength was assessed on the dominant hand using a hand-held dynamometer (Jamar handgrip dynamometer; Sammons Preston Roylan, Bolingbrook, IL, USA). The dynamometer handle grip width was individualized for each participant. Individuals performed the handgrip by holding their preferred hand in front of the shoulder, with the elbow bent at a 90° angle and instructed to squeeze the dynamometer as hard as possible for three seconds. Two attempts were allowed with the maximum force (kg's) recorded (*Reijnierse et al., 2017*). Participants were allowed a brief rest period (~1 min) between the two trials, with the maximum force produced across the two attempts utilised for statistical purposes.

### Gait speed

Gait speed was recorded using a computer interfaced electronic system (GaitMat II; EQ, Inc., Chalfont, PA, USA), which required participants to walk across a 3.66 m (11.91 ft.) long pressure mat system (*McDonough et al., 2001*). Participants completed two trials at their habitual gait speed in regular footwear. All participants were provided the following instructions: "Walk towards the end of the room in the center of the mat at a pace that is comfortable for you". All participants initiated from a standing start 2 m (6.56 ft.) from the gait analysis platform in order to minimize the effects that either acceleration or deceleration may have on the outcome measures (*Kressig et al., 2004*). Participants were permitted to have as much rest as was required between measures, with rest times typically being up to one minute. The fastest of each participant's two trials was kept for statistical analysis.

### Self-reported data

Participants in this study were also asked a series of questions regarding their number of years participating in Master sports, the number of hours training they would average in a typical training week as well as their number of chronic diseases and prescribed medications.

### Statistical analysis

All data was initially inspected for normality by investigating kurtosis, skewness, Q–Q plots, as well as the Kolmogorov–Smirnov test with the Lilliefors significance correction.

Heteroscedasticity was also assessed using Levene's test for the equality of variances. Overall group data were presented as means and standard deviations for continuous measures and counts (percentages) for categorical data. Between group comparisons were conducted on the basis of sex (male and female) and age (40–49 years; 50–59 years; 60–69 years; or $\geq$70 years) using one-way ANOVA and Tukey post-hoc tests. As many significant sex-related differences were observed and as there were some sex-related variation in the proportions of athletes across the age groups, data is presented for the entire sample as well as for females and males separately. A single sample $t$-test was performed to compare the height, mass and BMI of the Masters athletes (sex combined) to *Australian Bureau of Statistics (2012)*. All statistical analyses were performed with SPSS (Ver. 22.0.0.0), with statistical significance for all analyses accepted at $p < 0.05$.

## RESULTS

A total of 156 individuals 78 males (aged 40–86 years) and 78 females (aged 40–77 years) Pan Pacific Master Games sports athletes volunteered to participate in this study. One hundred and forty-nine individuals competed in one sport at the Games, with the remaining seven competing in multiple sports. With regard to chronic diseases, a total of 40 participants (25.4% of the overall sample) reported having one or more chronic diseases. Of these 40 participants, the most common chronic disease was hypertension ($n = 22$, 55%) and diabetes mellitus ($n = 14$, 35%). Of the 40 participants who answered having a chronic disease, it was reported that participants having at least one chronic disease as $n = 22$ (14%) whilst having two or more chronic diseases as $n = 18$ (11.6%). A total of 121 participants (77.6% of the overall sample) reported having no prescribed medications. A total of 18 participants reported having two or more chronic diseases, most commonly hypertension and dyslipidemia. The majority (77.6%) of the Masters athletes had no prescribed medications; with the majority of the remaining athletes ($\sim$11%) having one medication. There was no difference between sex with regard to the number of athletes being prescribed medications (males 16, females 19). The most commonly prescribed medications were anti-hypertension (55%) followed by oral hypoglycaemic agents (35%).

The anthropometric, functional, training and health descriptors of the overall sample is provided in Table 1. The only anthropometric variable where age-related differences were observed was height, where the 50–59 year-old athletes were significantly taller than the 60–69 year-old athletes ($p = 0.003$). From a functional perspective, there was a significant age-related decline in handgrip strength ($p = 0.003$) but no significant difference in habitual gait speed ($p = 0.673$). Based on our findings, Master athletes had a significant age-related increase in chronic diseases ($p = 0.017$) and prescribed medications ($p = 0.015$).

With regard to comparison to the Australian general population, as a group the Masters athletes were significantly taller ($+2.9$%, $p < 0.001$), slightly lighter ($-2.7$%) and a significantly lower BMI ($-8.5$%, $p < 0.001$). Figure 1 provides a representation of the proportion of individuals defined as underweight, normal weight, overweight or obese by sample and sex.

Due to their high level of lean muscle mass, no participants were described as being sarcopenic. However, the number of participants who were identified as having handgrip

**Table 1  Anthropometric, functional, training and health characteristics of total sample.**

| | 40–49 years (n = 42) | 50–59 years (n = 66) | 60–69 years (n = 30) | 70–79 years (n = 18) | Total (n = 156) |
|---|---|---|---|---|---|
| **Anthropometric characteristics** | | | | | |
| Height (cm) | 172.6 ± 8.4 | 173.6 ± 7.9[b] | 167.9 ± 11.0 | 171.1 ± 11.7 | 171.9 ± 9.3 |
| Mass (kg) | 76.6 ± 14.4 | 77.5 ± 15.8 | 73.3 ± 13.7 | 77.1 ± 17.7 | 76.4 ± 15.2 |
| Fat (%) | 22.9 ± 8.3 | 22.2 ± 9.1 | 26.9 ± 9.7 | 27.1 ± 10.3 | 23.8 ± 9.3 |
| Fat mass (kg) | 17.5 ± 7.3 | 17.6 ± 9.5 | 19.7 ± 8.7 | 21.2 ± 10.1 | 18.4 ± 8.9 |
| FFM (kg) | 59.1 ± 13.1 | 59.8 ± 11.3 | 53.5 ± 12.6 | 55.8 ± 13.7 | 57.9 ± 12.5 |
| BMI (kg/m$^2$) | 25.5 ± 3.5 | 25.6 ± 4.3 | 25.9 ± 4.7 | 26.3 ± 5.7 | 25.7 ± 4.3 |
| Sarcopenic status (count) | 0 | 0 | 0 | 0 | 0 |
| **Functional characteristics** | | | | | |
| Grip strength (kg) | 43.9 ± 13.2[b,c] | 43.8 ± 11.7[b,c] | 35.1 ± 12.4 | 31.9 ± 8.5 | 40.8 ± 12.7 |
| Gait speed (m/s) | 1.25 ± 0.18 | 1.28 ± 0.21 | 1.27 ± 0.19 | 1.26 ± 0.22 | 1.27 ± 0.20 |
| **Training and health characteristics** | | | | | |
| Participation (years) | 5.9 ± 5.9[a,c] | 10.4 ± 8.5 | 11.1 ± 10.6 | 15.8 ± 10.5 | 9.9 ± 9.0 |
| Training (h) | 6.0 ± 3.7 | 4.5 ± 3.3 | 4.3 ± 3.9 | 6.8 ± 5.1 | 5.1 ± 3.8 |
| Chronic diseases (number) | 0.1 ± 0.2[b,c] | 0.2 ± 0.6[b,c] | 0.8 ± 1 | 1.0 ± 1.0 | 0.4 ± 0.8 |
| Prescribed medications (number) | 0.1 ± 0.2[b,c] | 0.2 ± 0.7[c] | 0.6 ± 1.0 | 1.2 ± 1.2 | 0.4 ± 0.8 |

Notes.
  All values are mean ± SD except sarcopenic status whereby count is the number of individuals with sarcopenia in each category.
[a]Significantly different to 50–59 years old group.
[b]Significantly significant to 60–69 years old group.
[c]Significantly significant to 70–79 years old group.

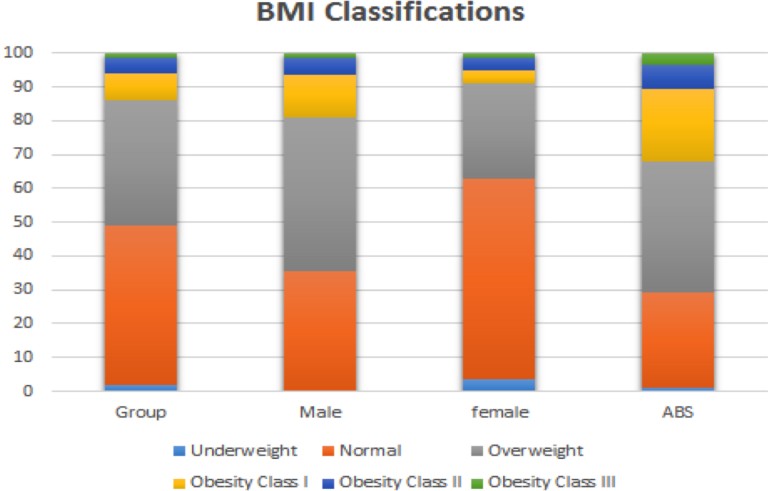

**Figure 1  Body Mass Index (BMI) classifications of overweight and obesity for the overall sample of Masters athletes as well as compared to *Australian Bureau of Statistics (2012)* data.**

muscle strength (n = 6) or physical performance outcomes (n = 1) below the sarcopenic cut-points (*Cruz-Jentoft et al., 2010*) are included in Fig. 2.

The anthropometric, functional, training and health descriptors of the 78 male Masters athletes are provided in Table 2. No significant age-related differences were observed for

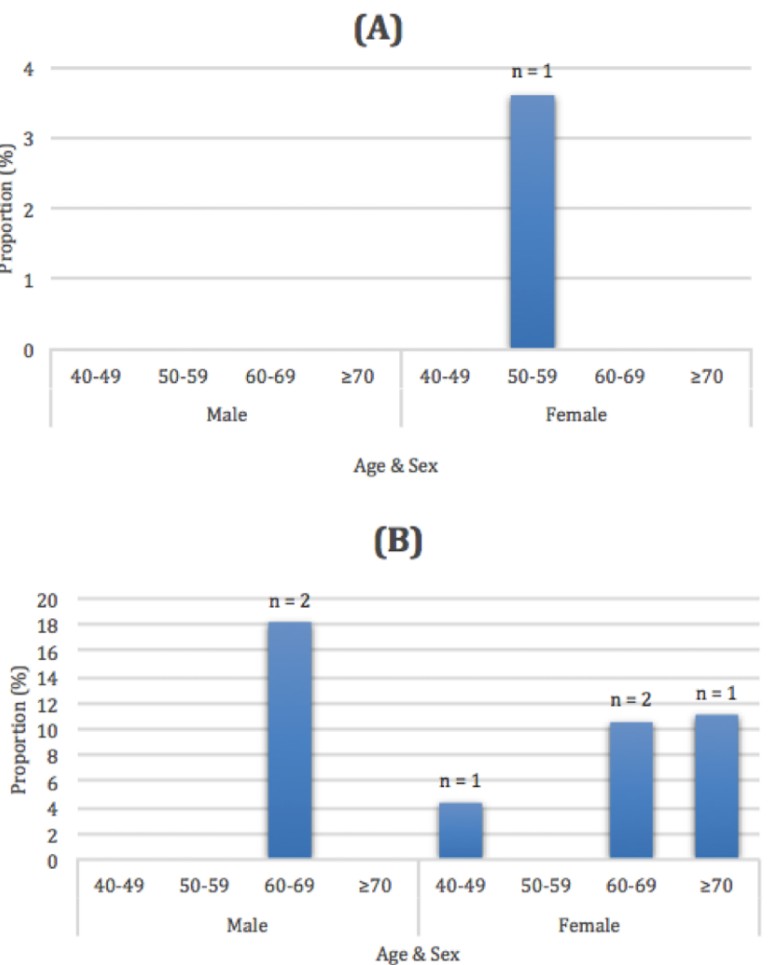

**Figure 2 Proportion and total number of Masters athletes who were below the EWGSOP gait speed (A) and handgrip strength thresholds (B) for the male and female Masters athletes across the 10 year age groups.** Gait speed (A) and handgrip strength thresholds (B).

any anthropometric variables. Inspection of the functional outcomes revealed a significant age-related decline in handgrip strength ($p < 0.001$) but no significant differences in habitual gait speed ($p = 0.930$). There was a significant age-related increase in chronic diseases ($p < 0.001$) and prescribed medications ($p < 0.001$).

The anthropometric, functional, training and health descriptors of the 78 female Masters athletes are provided in Table 3. There were typically no significant age-related differences observed for any anthropometric variables. The only exception was height which was found to be significantly greater in the 50–59 year than 60–69-year-old age group ($p = 0.003$). Inspection of the functional outcomes revealed a significant age-related decline in handgrip strength ($p = 0.003$) but no significant differences in habitual gait speed ($p = 0.673$). There was a significant age-related increase in chronic diseases ($p = 0.017$) and prescribed medications ($p = 0.015$).

**Table 2 Anthropometric, functional, training and health characteristics of male sample.**

| | 40–49 years (n = 19) | 50–59 years (n = 39) | 60–69 years (n = 11) | 70–79 years (n = 9) | Total (n = 78) |
|---|---|---|---|---|---|
| **Anthropometric characteristics** | | | | | |
| Height (cm) | 180.1 ± 6.2 | 176.7 ± 6.7 | 177.2 ± 12.6 | 178.3 ± 10.2 | 177.8 ± 8.0 |
| Mass (kg) | 88.3 ± 10.5 | 83.7 ± 15.7 | 83.3 ± 11.2 | 87.5 ± 12.7 | 85.2 ± 13.6 |
| Fat (%) | 18.2 ± 7.0 | 18.4 ± 8.2 | 20.3 ± 6.4 | 23.2 ± 5.6 | 19.2 ± 7.5 |
| Fat mass (kg) | 16.4 ± 7.5 | 16.4 ± 10.6 | 16.6 ± 4.8 | 20.6 ± 7.4 | 16.9 ± 8.9 |
| FFM (kg) | 71.9 ± 7.7 | 67.3 ± 7.4 | 66.7 ± 11.6 | 66.9 ± 8.7 | 68.3 ± 8.4 |
| BMI (kg/m$^2$) | 27.2 ± 3.1 | 26.8 ± 4.5 | 26.6 ± 3.2 | 27.4 ± 2.8 | 26.9 ± 3.8 |
| Sarcopenic status (count) | 0 | 0 | 0 | 0 | 0 |
| **Functional characteristics** | | | | | |
| Grip strength (kg) | 55.2 ± 9.4[b,c] | 51.3 ± 8.5[c] | 45.6 ± 14.0 | 37.5 ± 5.9 | 49.9 ± 10.7 |
| Gait speed (m/s) | 1.25 ± 0.19 | 1.25 ± 0.19 | 1.29 ± 0.14 | 1.24 ± 0.21 | 1.26 ± 0.18 |
| **Training and health characteristics** | | | | | |
| Participation (years) | 6.2 ± 6.3[c] | 12.6 ± 9.7 | 14.2 ± 11.0 | 19.6 ± 12.8 | 12.1 ± 10.2 |
| Training (h) | 6.6 ± 4.2 | 4.1 ± 3.6 | 5.3 ± 4.3 | 7.2 ± 6 | 5.2 ± 4.3 |
| Chronic diseases (number) | 0.0 ± 0.0[b,c] | 0.1 ± 0.3[b,c] | 1.2 ± 1.0 | 1.0 ± 0.9 | 0.3 ± 0.7 |
| Prescribed medications (number) | 0.1 ± 0.2[b,c] | 0.1 ± 0.3[b,c] | 0.9 ± 1.2 | 1.1 ± 0.9 | 0.3 ± 0.7 |

**Notes.**
All values are mean ± SD except sarcopenic status whereby count is the number of individuals with sarcopenia in each category.
[a]Significantly different to 50–59 years old group.
[b]Significantly significant to 60–69 years old group.
[c]Significantly significant to 70–79 years old group.

# DISCUSSION

The primary results of this study indicated that the male and female Masters athletes who participated in this study possessed substantially better anthropometric, functional and health characteristics than the generally less active Australian population their physically less active age-matched peers. In 2011–12, adults spent an average of 33 min per day doing physical activity, yet the distribution was highly skewed with 60% of adults doing less than 30 min, and fewer than 20% doing an hour or more per day on average. In contrast, sedentary leisure occupied just over 4 h a day on average, with almost 30% of the adult population reporting more than 5 h of sedentary leisure activity each day. Our data indicated that when the anthropometric, functional and health characteristics were compared for the Masters athletes across the 10 year age groups, almost no significant age-related differences were observed. In contrast, our data indicated significant age-related reductions in handgrip strength and increases in the number of chronic disease and prescribed medications.

When compared to the *Australian Bureau of Statistics (2012)* BMI data, it was apparent that Masters athletes had lower BMI which is suggestive of a healthier body composition. In particular, while only 29% of the general population have a BMI indicative of underweight or normal, approximately 49% of the Masters athletes were in these categories. Further, while ~32% of the Australian adult population was classified as obese (BMI ≥ 30.0 kg/m$^2$), only ~13% of the Masters athletes were obese. Such results were consistent with previous

**Table 3  Anthropometric, functional, training and health characteristics of female sample.**

| | 40–49 years (n = 23) | 50–59 years (n = 27) | 60–69 years (n = 19) | 70–79 years (n = 9) | Total (n = 78) |
|---|---|---|---|---|---|
| **Anthropometric characteristics** | | | | | |
| Height (cm) | 166.4 ± 3.9 | 169.1 ± 7.1[b] | 162.6 ± 5 | 164 ± 8.6 | 166.1 ± 6.5 |
| Mass (kg) | 67.2 ± 9.9 | 68.6 ± 11 | 67.5 ± 11.7 | 66.6 ± 16.1 | 67.7 ± 11.3 |
| Body Fat (%) | 27.4 ± 6.7 | 27.7 ± 7.5 | 30.7 ± 9.4 | 31.0 ± 12.6 | 28.7 ± 8.4 |
| Fat mass (kg) | 18.9 ± 7 | 19.5 ± 7.5 | 21.6 ± 10 | 21.9 ± 12.7 | 20.1 ± 8.6 |
| FFM (kg) | 48.3 ± 4.3 | 49 ± 5.7 | 45.9 ± 3.8 | 44.7 ± 6.8 | 47.6 ± 5.2 |
| BMI (kg/m$^2$) | 24.3 ± 3.3 | 24 ± 3.7 | 25.7 ± 5.4 | 25.2 ± 7.6 | 24.6 ± 4.6 |
| Sarcopenic status (count) | 0 | 0 | 0 | 0 | 0 |
| **Functional characteristics** | | | | | |
| Grip strength (kg) | 34.1 ± 6.4[b,c] | 33 ± 5.3[c] | 29.2 ± 5.9 | 26.4 ± 7.0 | 31.6 ± 6.5 |
| Gait speed (m/s) | 1.24 ± 0.17 | 1.31 ± 0.23 | 1.26 ± 0.20 | 1.27 ± 0.23 | 1.27 ± 0.21 |
| **Training and Health Characteristics** | | | | | |
| Participation (years) | 5.7 ± 6.0 | 7.3 ± 5.2 | 9.4 ± 10.2 | 12.1 ± 6.3 | 7.9 ± 7.2 |
| Training (h) | 5.4 ± 3.2 | 5.2 ± 2.8 | 3.8 ± 6.7 | 6.4 ± 4.4 | 5.1 ± 3.4 |
| Chronic diseases (number) | 0.1 ± 0.3[c] | 0.5 ± 0.8 | 0.6 ± 1.1 | 1.1 ± 1.2 | 0.5 ± 0.9 |
| Prescribed medications (number) | 0.1 ± 0.3[c] | 0.5 ± 1.1 | 0.5 ± 0.9 | 1.3 ± 1.5 | 0.5 ± 1 |

Notes.

All values are mean ± SD except sarcopenic status whereby count is the number of individuals with sarcopenia in each category.

[a]Significantly different to 50–59 years old group.

[b]Significantly significant to 60–69 years old group.

[c]Significantly significant to 70–79 years old group.

research that demonstrated that a high proportion of Masters athletes maintain a healthy BMI even into older age (*Climstein et al., in press*; *Walsh et al., 2011*; *Walsh et al., 2013b*). As obesity is a well-recognized risk factor for many chronic diseases including hypertension (resting blood pressure ≥ 140/90 mmHg), cardiovascular disease, type 2 diabetes mellitus, obstructive sleep apnea and a number of cancers (*Lavie, Milani & Ventura, 2009*; *Pi-Sunyer, 2009*), based upon our findings, it appears that long-term Masters sports participation results in a lower (improved) BMI and subsequently this reduces their risk for many chronic diseases (*Climstein et al., in press*; *McKean, Manson & Stanish, 2006*; *Pratley et al., 1995*).

Subgroup analyses within the current study indicated that the differences in BMI and obesity rates between the Masters athletes and the wider Australian population were more pronounced for females than males. Specifically, ∼63% of the females compared to ∼35% male Masters athletes were classified as being underweight or normal weight; with only ∼8% of female and 19% of male Masters athletes being obese. This tendency for the female Master athletes to report lower BMI measures than male Master athletes in terms of their BMI and obesity rates appeared consistent with *Climstein and colleagues (in press)* for the rates of chronic disease. Such results may suggest that middle-aged and older females may obtain significantly greater body composition and chronic disease benefits from participation in Masters sport than men of similar age.

Another important aspect of maintaining health and function in ageing is the ability to maintain muscle mass. None of the 114 Masters athletes aged 50 years and older (48

of who were 60 years or older) in the current study were diagnosed as having sarcopenia, based upon the EWGSOP criteria. Such prevalence rates for sarcopenia in the current study are substantially lower than the 1–29% rates reported for community dwelling adults over the age of 50 years in a systematic review (*Cruz-Jentoft et al., 2014*). Inspection of the other components of the sarcopenia diagnosis (muscle strength and physical performance) indicated that only 3.8% of the participants were below the sarcopenic handgrip strength cutpoints and 0.6% were below the sarcopenic gait speed cutpoints. As ~85% and ~97% of Australian aged care residents may be below the sarcopenic handgrip strength and gait speed thresholds, respectively (*Keogh et al., 2015*), our data suggest that Master athletes levels of muscular strength and physical function place them at very low risk of adverse health events and entry into residential aged care (*Abellan van Kan et al., 2009*).

Results of the current study also indicated a very low prevalence of chronic diseases and medications in the Masters athletes. Such a result was consistent with *Climstein and colleagues (in press)* who observed a low rate of cardiovascular disease in Masters athletes. While the current study observed a significant age-related increase in the number of chronic diseases and prescribed medications, even those Masters athletes who were 70 years old or older only reported an average of one chronic disease and one prescribed medication per individual. This number of chronic diseases and medications was consistent with *Climstein and colleagues (in press)* and is substantially less than community dwelling older adults of similar age, especially those in residential aged care (*Olsen et al., 2016*). This is exemplified by recent Australian data in which older adults living in residential aged care may average 14–15 chronic diseases and 11–15 prescribed medications per individual (*Fien et al., 2016*; *Keogh et al., 2015*).

The results of this current study and the remainder of the somewhat limited literature regarding Masters athletes strongly support the promotion of physical activity and sport across the lifespan. This promotion of a lifetime engagement in physical activity and sport may provide middle-aged and older individuals as well as the wider community a host of benefits. Physical activity and sport participation can have a variety of physiological benefits that reduce or even reverse the age-related declines in muscle mass, strength and performance (*Fien et al., 2016*; *Kanamori et al., 2012*). Sports participation also provides a range of social benefits (*Kanamori et al., 2012*), with this especially important for older adults who may experience declines in the size and quality of their social networks that accelerated when they retire from full-time employment (*Julien et al., 2013*). From an economic perspective, older adults typically utilise a very high proportion of most nations' health care expenditure due to the number of their chronic diseases, prescribed medications and falls-related injuries (*Manini & Pahor, 2009*). Of particular concern is that the 20% of older adults who are no longer independent due to functional limitations have been shown to account for 46% of the health care expenditure (*Manini & Pahor, 2009*). In absolute terms, this represents a $5,000 per year greater healthcare expenditure for each older adult with functional limitations compared to those who have maintained their physical independence (*Manini & Pahor, 2009*). Such savings in healthcare expenditure compared to dependent older adults, may be even greater in Masters athletes than that reported for independent older adults. Specifically, data from the present study indicating that the 48
Masters athletes over the age of 60 years typically averaged only one chronic disease and one prescribed medication. This contrasts with Australian data in which over half of Australian adults aged 65 years and older have at least five chronic diseases (*Australian Institute of Health and Welfare, 2014*).

The primary strength of the study was the sample size of 156 Masters athletes. While a few studies of Masters athletes have had sample sizes that exceeded 400 participants (*Climstein et al., in press*; *Walsh et al., 2011*; *Walsh et al., 2013a*; *Walsh et al., 2013b*), all of these studies with the exception of *Climstein and colleagues (in press)* only collected BMI and used this to estimate obesity rates. In comparison, Masters athlete studies that involved a greater number of outcome measures or utilised more advanced data collection approaches have typically obtained much smaller sample sizes ($n = 6$–40) than that used in our study (*Gacesa, 2017*; *Pantoja et al., 2016*; *Power et al., 2016*; *Pratley et al., 1995*; *Wroblewski et al., 2011*; *Young et al., 2008*).

The major limitation of this study was that the study was cross-sectional rather than longitudinal in design. This meant that while age-related differences between the age groups could be described, true age-related changes that would be observed in individuals as they age could not be quantified (*Young et al., 2008*). Nevertheless, the data did indicate the favorable anthropometry, functional and health characteristics of the Masters athletes compared to data reported for less physically active, age-matched individuals. There was also some imbalance in the male and female ratios for 50–59 and 60–69 year old age groups which may have contaminated the overall sample data. Age-related analyses for the 78 male and 78 female Masters athletes were therefore performed separately in addition to data presented for the overall cohort of 156 athletes. It is also unclear how the athletes who participated in the study may differ to the athletes who did not participate in the study and if there was any recruitment bias whereby the more functional and healthy Masters athletes would be more likely to participate than their less functional or healthy peers.

## CONCLUSIONS

The results of this study indicate that Masters athletes possess more favourable anthropometric, functional and health characteristics than the age-matched general community who are typically much less physically active. Further, comparisons of the decade age groups also indicated that the majority of these characteristics were well maintained from middle to older age in the Masters athletes. Such data supports the promotion of a lifespan physical activity and sport public health message, with such participation having a range of physical, social and economic benefits for the individual and the wider community. Future research in this area should look to include a wider range of health and functional outcomes and utilize longitudinal case-control research designs to better compare the age-related changes in these outcomes between Masters Athletes and their less active age-matched peers.

## ACKNOWLEDGEMENTS

We would like to thank the organisers and athletes of the 2014 Pan Pacific Masters Games for their assistance in conducting the study. We would also like to thank the Bachelor of Exercise and Sport Science students from the Faculty of Health Sciences and Medicine, Bond University who helped to collect the data.

### Funding
The authors received no funding for this work.

### Competing Interests
Timothy Henwood is the Group Manager, Connected Living—Community Wellness and Lifestyle of Southern Cross Care (SA & NT) Inc., Adelaide, South Australia, Australia. Justin W.L. Keogh is an Academic Editor for PeerJ.

### Author Contributions
- Samantha Fien conceived and designed the experiments, performed the experiments, analyzed the data, contributed reagents/materials/analysis tools, wrote the paper, reviewed drafts of the paper.
- Mike Climstein and Justin W.L. Keogh conceived and designed the experiments, analyzed the data, contributed reagents/materials/analysis tools, wrote the paper, prepared figures and/or tables, reviewed drafts of the paper.
- Clodagh Quilter and Georgina Buckley analyzed the data, contributed reagents/materials/analysis tools, wrote the paper, prepared figures and/or tables, reviewed drafts of the paper.
- Timothy Henwood conceived and designed the experiments, reviewed drafts of the paper.
- Josie Grigg conceived and designed the experiments, performed the experiments, reviewed drafts of the paper.

### Human Ethics
The following information was supplied relating to ethical approvals (i.e., approving body and any reference numbers):

This study received approval by the Bond University Human Research Ethics Committee (RO1823) in accordance with the ethical standards of the Helsinki Declaration of 1975 (revised in 2008).

### Data Availability
The raw data has been uploaded as Supplemental Files.

## Supplemental Information

Supplemental information for this article can be found online at http://dx.doi.org/10.7717/peerj.3768#supplemental-information.

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
