# Peer review of "Anthropometric, physical function and general health markers of Masters athletes: a cross-sectional study"

_PeerJ, doi:10.7717/peerj.3768_

## Round 0.1 · original submission · Minor Revisions

Thank you for your submission. It was a pleasure to read this manuscript. Please review the comments and the suggestions from each of the reviewers, including the annotated manuscript. Upon resubmission, please address each comment in your response and highlight changes to the manuscript. I look forward to the next version.
Scotty

·

Basic reporting

The article as a whole is well written using clear and ambiguous language. Although there are some concepts that are "confusing" these do not seem to be purposeful and may be simply author's vast knowledge and involvement with the manuscript.

As a whole, the topic seems well research and sufficient background and context is provided for the reader to understand.

Overall, the manuscript is well written and provides significant contribution to the pertinent field of study. However, there are several areas where the manuscript could use some clarity.

Experimental design

Overall the study design is robust and provides meaningful data. The research question is well defined and the authors provide adequate support of their aims, objectives and statistical methods to reach appropriate conclusions.

I have provided additional comments throughout the text, which may require some additional information to add clarity.

Validity of the findings

The results provided seem to be valid and reliable. However, the results section as described by the authors requires some work to improve clarity for the reader.

I would encourage authors to address some of the comments provide to add clarity to some of these points.

Additional comments

I commend all the authors for their work in this topic. Although the work requires some additional work, the findings as described provide good insight on the benefits of engaging in Sports late in life.

Reviewer 2 ·

Basic reporting

This manuscript appears to meets all the requirements as set by PeerJ policies and the reporting in accordance with STROBE guidelines.
The introduction is too long. This is not a review of the literature. A shorter, more focused introduction to the study would be better. Much of the literature presented could be synthesised with one or two relevant supporting citations.
There are some issues with punctuation that need to be addressed. The authors need to ensure punctuation throughout the document is correct.

Experimental design

Is the BIA instrument used in the study the same as that used for the Australian reference population? There are no cut points or thresholds for any of the outcome measures. How was sarcopenia evaluated, ie using height adjusted measures, relative or absolute measures of skeletal muscle mass, or fat free mass? What are the expected thresholds for the ‘sarcopenic gait speed’? What are the reference values for handgrip?

Validity of the findings

In the Results, I am not sure why a percentage for differences in BMI is used with the reference population? Could the kg/m2 differences be included?

The discussion was acceptable, but the authors state in line 235 that 49% of the cohort were ‘underweight or had normal body composition’. What percentage were underweight?

---

## Round 0.2 · accepted · Accept

Congratulations on your acceptance and thank you for your well written manuscript.
Scotty

·

Basic reporting

As before, this article is well written and the authors demonstrated a vast knowledge of the topic. The authors have address the previous concerns and the article seems to read more clearly.

Experimental design

I have no additional comments from my original review. The authors have address my concerns throughout the manuscript.

Validity of the findings

The authors have address my concerns with this section.

Additional comments

I commend the authors for their work in the area of research.

Reviewer 2 ·

Basic reporting

The authors have adequately addressed all issues raised by the reviewers.

Experimental design

The authors have adequately addressed all issues raised by the reviewers.

Validity of the findings

The authors have adequately addressed all issues raised by the reviewers.